# Rank-Related Differences in Dogs’ Behaviours in Frustrating Situations

**DOI:** 10.3390/ani14233411

**Published:** 2024-11-26

**Authors:** Kata Vékony, Viktória Bakos, Péter Pongrácz

**Affiliations:** 1Department of Ethology, ELTE Eötvös Loránd University, Pázmány Péter sétány 1/c, 1117 Budapest, Hungary; kata.vekony.kami@gmail.com (K.V.);; 2Institute of Cognitive Neuroscience and Psychology, HUN-REN Research Centre for Natural Sciences, Magyar Tudósok krt. 2, 1117 Budapest, Hungary

**Keywords:** dog, hierarchy, rank, frustration, social context, non-social context

## Abstract

Frustration is a negative emotion elicited by situations when a positive expectation of an individual is violated, for example, when an anticipated reward is delayed or withheld, or a previously accessible resource becomes inaccessible. We investigated whether social rank can be a factor in the individual variability among dogs’ tolerance to frustrating situations. We used a validated questionnaire to assess the ranks of dogs from multi-dog households and measured their behaviour in a social and a non-social frustration test, where a previously accessible treat became inaccessible either because the experimenter withheld it, or it was in a closed cage. All dogs’ behaviours changed between the two tests; however, dominant dogs showed more dependence on their owner in the non-social test than in the social one, while subordinate dogs’ owner-directed behaviours did not change. More agonistic dogs were more persistent in the non-social test, while dogs that tend to be the ‘leaders’ in everyday contexts acted more demanding towards the experimenter when she kept the reward away from the dog. Our results show that the social rank of dogs affects their resource-related behaviours in frustrating situations, although the relevance of different aspects of rank depends on the social context.

## 1. Introduction

Dogs are not only the oldest domesticated species (e.g., [1]), but they also show a near-perfect adaptation to the anthropogenic environment, which is considered as being the natural habitat for dogs [2]. Consequently, the vast majority of the world’s one billion-strong dog population [3] habitually and continuously lives together with humans, whether belonging to free-ranging dogs [4] or representing a companion animal [5]. One of the main characteristics of dogs is a complex system of dependence on humans, both with regard to the procurement of vital resources [6] and emotional–behavioural bonding [7] that creates an asymmetrical relationship between the dog and usually one specific person (who is usually referred to as the owner/main caretaker). There are several specific socio-cognitive traits that ensure and maintain this close asymmetric tie between the dog and its focal person; among these are dog–owner attachment [8], mutual emotional and communicative understanding [9], and reliance on human assistance in difficult situations [10,11,12,13], which is of primary importance here.

Dogs are not merely well-trainable helping assets to humans, but they also seem to have developed a specific willingness to cooperate with us in joint actions, preferring human partners over their conspecifics with whom they rather maintain a competitive/neutral relationship [14]. Dogs (from this point on, we always refer to dogs which have an owner/handler) can show strong variability in their reliance on human-provided cueing and assistance depending on their genetic distance from the common ancestor [15], breed-specific selection [13], or level of training [16]. However, in general, dogs still show a more prevalent dependency on human contribution in difficult-to-solve situations than (socialised) wolves do [12]. However, this dependency comes with a cost: when dogs face unsolvable problems and they do not receive the usual assistance from their human partner, they are prone to various levels of stress, which, depending on the context, can manifest in problematic behaviours (e.g., separation-related problems [17]). When dogs previously learned that a positively valenced (rewarding) event would come, which in turn is delayed or cancelled, they can react with frustration. Frustration is an aversive emotional state [18,19] that can arise in a range of situations [20], such as when an expected reward is absent, delayed [18], reduced in value [21], or inaccessible due to barriers [22], which, importantly, can be of a physical or social nature.

Dogs’ reaction to a delayed, blocked, or absent positive stimulus (or resource) can depend on their previous experiences with the situation (the more they know how to operate an unanimated food-dispenser device successfully, the fewer stress signs they show when the device stops providing the reward [23]). When dogs are separated from their caretaker (i.e., they are left at home alone), they lose access to their single most crucial resource, and this is thought to be a main contributor to separation-related problems. It was found that dogs that reacted with frequent barking to a short separation episode mostly lived with ‘lenient owners’, who normally easily give in when the dog behaves demandingly [24]. However, there was a strong variability in dogs’ responses to frustration-evoking contexts even when the researchers used rather similar stimuli. For example, when an experimenter at first provided a reward to the dogs but then refrained from this, the dogs reacted by keeping a larger distance from the human, and either inactivity [25] or increased general ambulation, vocalisation, and sniffing around followed [19]. However, Kuhne [26] reported that when the experimenter stopped rewarding the subjects and held the treat in her hand, the dogs either tried to obtain the reward by manipulating the experimenter’s hand or stared motionlessly at her.

But what can cause differences in dogs’ behavioural and emotional reactions to frustrating situations? Breed type was found to be a strong factor as representatives of cooperative working dog breeds reacted stronger to the absence of their owner [27] and showed more human-directed behaviours in a difficult-to-solve task [13]. McPeake et al. [20] developed and validated [28] the Canine Frustration Questionnaire, and they found that dogs show stable traits connected to their frustration-related reactions. Lifetime experiences, such as participating in training activities that support dogs’ resilience against frustration (e.g., ‘obedience’ training, [29]) or enhance their persistence in situations of difficult-to-solve problems [30], can also influence the behaviour of dogs in frustrating situations.

Another important feature that can affect dogs’ responses to difficult-to-handle situations is their rank in the social hierarchy. The position of an individual within the hierarchy develops through social interactions [31] and may even change dynamically during the life of a dog [32]. Following the well-established ethological definitions [33], a ‘dominant dog’ is an individual who regularly prevails over the other cohabitant or group member dogs (the ‘subordinate’) in dyadic competitive events and, as a consequence, gains priority to resources. There are indications that among dogs, a well-established hierarchy is mostly maintained by signals of submission from the subordinate dogs rather than by repeated displays or acts of aggression [32]. Personality traits (e.g., extraversion, openness, conscientiousness, agreeableness) that show a non-causative association with the rank position of an individual dog [34] may also be influential in their reactions to frustration-generating situations.

While the position of an individual in the hierarchy can mainly be traced back to resource-related interactions, social rank has other aspects and presentations that are not strictly tied to resource competition. In one recent paper, a complex approach was used for social ranking, which included three distinct aspects [31]. The association of resource possession and greeting behaviour with formal, agonistic, and leadership rank scores in cohabiting dogs was investigated by Vékony and Pongrácz [31], and the results indicated that the various rank components’ effects may depend, to a different extent, on the dog’s social skills. This suggests that they may also have different relevance in social and non-social situations. ***Formal rank*** can be described as directionally consistent affiliative displays between dominant and subordinate individuals [35]. Formal rank was originally thought to be a direct extension of the established ***agonistic rank***—but this latter rank-type is determined by competitive behaviours. Research on free-ranging dogs found that not every individual consistently receives submissive displays in agonistic contexts from the others but also receives formal submission (outside of direct competition). Usually, younger individuals with high agonistic rank do not possess the same formal rank, suggesting that formal rank requires social skills more than competitive successes [36]. A third aspect that is related to but does not necessarily coincide completely with the previous two is ‘***leadership***’, which was found to be related more to affiliative relationships and formal rank in the case of group movement leadership [36] but only to some extent and in certain cases of leading territorial defence [37]. Although companion dogs’ environment is much more relaxed in terms of the need for resource competition or territorial defence, and much more controlled (by humans) in terms of choosing group-mates and almost all aspects of life, agonistic and formal-affiliative displays [31,38] and the leading of group movements [39] were also described in both temporary and cohabiting dog groups.

In this study, we tested higher- and lower-ranking cohabitant companion dogs, whose position in the hierarchy we assessed with the help of a validated questionnaire, which measures not only rank as a whole but also separately measures the aforementioned three aspects as well (Dog Rank Assessment Questionnaire or ‘DRA-Q’ [31]). The main aim of this study was to investigate the possible differences in low- and high-ranking and cohabiting dogs’ behavioural responses to such contexts that may elicit frustration. We hypothesised that there might be a two-way association between social rank and frustration tolerance. We assumed that individuals with higher motivation and ability to obtain and keep resources could end up as being dominant over individuals with lower motivations for possessing resources. Dominant individuals usually gain more positive experiences from obtaining desired resources; thus, consequently, we predicted that the higher-ranking dogs would show less tolerance (i.e., stronger frustration) for situations where the resource suddenly becomes inaccessible.

The second aim was to assess whether the social nature of the context (i.e., the source of frustration can be connected to an unanimated obstacle vs. a non-cooperative human) would differently affect the responses of higher- and lower-ranking dogs and the role the different aspects of social rank would play in each context. We predicted that in the non-social context, overall rank would not affect dogs’ persistence to obtain the reward or target-directed behaviours, but more agonistic dogs might be more focused on the reward, because their rank was affected stronger by their capacity to win resource-related contests. We predicted that dominant dogs are more dependent on their owner; thus, we expected that the overall higher-ranking dogs might show more dependent behaviours towards the nearby owner even in the case of the non-social problem task. This dependency might extend to humans in general [40,41], so we also expected to see more experimenter-directed behaviours, especially in the social context. As the leadership aspect of rank is strongly related to social skills, we predicted that it would have the strongest effect in the social context, with higher-ranking dogs presenting more experimenter-directed, communicative behaviours.

## 2. Methods

### 2.1. Subjects

Companion dogs over 1 years of age from multi-dog households participated in the tests (N_households_ = 18, N_dogs_ = 39, M_age_ ± SD = 6.66 ± 3.16, from 18 breeds and also mixed breeds, 17 females, 22 males, 37 spayed/neutered, Appendix A). The ranks, rank scores, and three subscores (agonistic, formal, and leadership) of the participants were determined by the DRA-Q [31] completed by the dog owners (distribution of ranks: N_dominant_ = 17; N_subordinate_ = 18; N_flexible_ = 2). Participation was voluntary, and all owners signed an informed consent form. We had to exclude one subject from the analysis of the first test because of owner non-compliance with the testing protocol.

All tests took place in the laboratory (6.27 × 5.4 m testing room) of the ELTE Department of Ethology, Budapest, Hungary.

### 2.2. Non-Social Frustration Test (Based on [42])

The experimental setup of the non-social frustration test is shown in Figure 1. A large metal dog crate was set up at the wall of the room with a plate in it; the experimenter and the owner stood approx. 2 m from the cage and 1.5 m from each other.

#### 2.2.1. Training Phase

Both the experimenter (E) and owner (O) stood at the designated starting point; the dog (D) was held on leash by the O. The E called the D’s attention (name of the dog + ‘Look!’), showed them a piece of a treat (sausage) then placed it on the plate in the cage, and then walked back to the starting point, leaving the door of the cage open. The O took off the leash, and the D was free to go and obtain the treat. This phase consisted of four trials. During each trial, the E was standing at her starting point.

#### 2.2.2. Test Phase

After the training phase, the O and D left the room. The E put 10 pieces of treats on the plate and placed the plate in the cage, then closed its door. The O and D were called back to the testing room, and when they arrived back to their starting point, the O let the D go, and the D was free to try to obtain the reward and also freely move around for 90 s. In this phase, the D was unable to obtain the treats from the closed cage. The O and E stood silent at their spots during the test (i.e., neither of them were allowed to encourage the dog to obtain the food).

### 2.3. Social Frustration Test (Based on [19,43,44])

During the test, the E sat on a chair in the middle of the room, while the O stood in a corner next to the door, and the dog was off leash (Figure 2).

#### 2.3.1. Training Phase

The E sat on the chair and gave two pieces of treats to the D, then held the food in her closed hand and raised her arm to the side, somewhat higher than horizontal out of the D’s reach. During the next two minutes following the provision of the first treats, the E praised the D and gave them a treat each time the D established eye contact with the E. Between two eye contact events, the E returned her hand back to the side.

#### 2.3.2. Test Phase

After two minutes of the training phase passed, the E ceased praising the D and giving them treats; in vain, the D established eye contact with her. The E kept her arm and hand stretched out to the side, so the dog was not able to obtain any treats. This phase lasted three additional minutes.

### 2.4. Behavioural Coding

Both tests were recorded using a six-camera system mounted to the ceiling of the testing room (cameras: Basler a2A1920-51gcPRO-Basler ace 2, Basler AG, Ahrensburg, Germany; microphone: Sennheiser ME-64 + K6-P powermodule, Sennheiser Electronic GmbH & Co., Wedemark, Germany; audio interface: Focusrite-Scarlett 2i2, Focusrite PLC, Wycombe, UK), and we coded the video footage using Solomon coder (beta 17.03.22 © András Péter, free software, (RRID:SCR_016041). Table 1 shows the coded behavioural variables. A total of 10% of the videos were coded by a blind coder for reliability testing.

### 2.5. Statistical Analyses

Statistical analyses were performed with R statistical software (v4.4.1, R Core Team, 2023) in RStudio (Build 764, © Posit Software, PBC), with the packages caret, corrplot, DataExplorer, dplyr, emmeans, fitdistrplus, glmmTMB, GPArotation, irr, lme4, lmerTest, paran, performance, psych, and rstatix. Raw data are included in Appendix A. We calculated the ICC for the behaviours to ensure inter-rater reliability. We used the Wilcoxon Signed Rank test to compare the behaviours of the dogs between the two tests and also to compare dominant versus subordinate dogs from the same household. As cage-related behaviours only occurred in the non-social test (no cage was present in the social test), we compared these to the experimenter-related behaviours of the social test, as the experimenter held the treat. We used the Benjamini–Hochberg correction to reduce the false discovery rate, and here, we report the adjusted *p* values. We performed Principal Component Analysis (PCA) on the behaviours in the two tests separately. The resulting components were then used as dependent variables. We used GLMMs with the household as the random effect; PCs as dependent variables; and rank scores obtained from the questionnaire, age, and sex as predictors to see if the holistic rank score or the more detailed agonistic and leadership scores affect frustration behaviours in either test. The formal score was excluded from the analysis, because many dog owners answered with “I don’t know” to the single question that refers to this aspect of rank. We used AIC-based model selection to find the most parsimonious model.

## 3. Results

The ICC showed excellent reliability (>0.9) for most behaviours in the non-social test; “Moving” and “Standing” had moderate reliability (<0.75). Most behaviours had good to excellent reliability in the social test, and “Vocalisation”, “Standing”, “Whining”, and “Barking” had moderate reliability, while “Other” had poor reliability (0.42).

Some of the behavioural variables had only sporadic occurrences; these we report descriptively. Gaze alteration between the reward and the owner was extremely rare: it occurred in the case of only one dog in the non-social test and two dogs in the social test. Yawning was similarly rare with one and two dogs yawning in the two tests, respectively. Dogs barely made physical contact with the owner in the two conditions: only one dog contacted the owner in the non-social test and four in the social test. Only one dog barked in the non-social test, while barking was more common in the social test. As for the various vocalisation types, especially “Barking” and “Other” were rare; we grouped these variables together into a “Vocalisation” variable, which included “Whining”, “Barking”, and “Other”.

When we analysed the behaviour of all dogs together, regardless of their rank, the Wilcoxon test revealed significant differences in most behaviours between the two tests. Table 2 shows the results.

The analysis of the effect of test context in dominant and subordinate dogs separately revealed that the results in the case of all dogs cannot be generalised to the dominant and subordinate dogs (Table 3). Differences in reward- and human-directed behaviours are shown in Figure 3.

The PCA on the behaviours in the non-social test revealed five principal components, which we named Determined, Hopeful, Complaining, Passive, and Begging. Together, they explained 78% of the total variance. Behaviours in the social test provided five principal components that we named Demanding, Complaining, Focused, Passive, and Excited. After further analysis, the component Excited was found to be inconsistent (Cronbach’s α < 0.3), so we excluded it from the analysis. Together, these components explained 67.3% of the total variance. The loading of the individual behaviours within each component and the principal components’ Cronbach’s α values are shown in Table 4.

We found no association between the principal components and the holistic rank score in either of the tests. Using the agonistic and leadership subscores separately in the model selection, we found a significant association between the agonistic score and the determined PC: dogs with higher agonistic scores were more determined in the non-social test (β = 0.4297, SE = 0.17, *t* = 2.563, 95%CI = (0.1011–0.7583), *p* = 0.0104, Figure 4A). The only other association in the non-social test was between age and begging: older dogs begged the experimenter less (β = −0.1495, SE = 0.07, *t* = −2.028, 95%CI = (−0.2939–−0.0050), *p* = 0.0425, Figure 4B).

In the social test, we found a significant positive association between the leadership score and “Demanding” (β = 0.3945, SE = 0.11, *t* = 3.514, 95%CI = (0.1745–0.6145), *p* = 0.0004, Figure 5A). Another significant association we found was between “Age” and “Focused”, older dogs being less focused on the experimenter than younger ones (β = −2.3546, SE = 0.76, *t* = −3.115, 95%CI = (−3.8334–−0.8758), *p* = 0.0037, Figure 5B). We found no other associations in the social test.

## 4. Discussion

In this study, we investigated how the social rank of cohabiting family dogs, measured by a validated instrument, affects their actual behaviours in frustration-eliciting situations in both non-social and social contexts. We found that dogs’ behaviour was different in the two contexts regardless of their rank, but this change might be different for dominant and subordinate dogs.

All dogs were more active when the reward was inaccessible in a closed cage, suggesting that they either tried more persistently to obtain the reward on their own, or they might have become more aroused and stressed in the non-social context. As dogs spent more time sitting in the social test, this suggests that including a human as a social partner can reduce independent attempts at reaching the reward, highlighting dogs’ social dependence. The more passive behaviour of the dogs in the social scenario may also show the effect of past experience (or training), in which dogs could have learned that people do not like strongly demanding reactions from frustrated dogs. On the other hand, when the reward became inaccessible in the cage, at least initially, dogs could see this scenario as a problem-solving task which they should solve on their own.

An interesting finding was how dogs with different ranks in the case of some (but not each) of the behaviours differentiated between the non-social and social contexts. Both subordinate and dominant dogs clearly acted differently towards the experimenter and the cage when the reward became inaccessible (i.e., they all spent more time near the experimenter than in the vicinity of the cage). However, subordinate dogs spent more time gazing at the closed cage in the first test than at the experimenter who withheld the reward in the second test, suggesting that they would rather try and solve the problem without asking for a stranger’s help. This result shows parallels with earlier publications, where in a social learning context, it was found that the subordinate dogs benefitted less from observing an unfamiliar human demonstrator than the dominant dogs did [40,41]. When it comes to communication with the owner though, dominant dogs relied more on their well-known social partners: in the non-social test, they gazed more towards the owner than in the social test, while subordinate dogs did not seem to find the owner’s potential involvement in the two situations to be different. These results provide an intriguing insight into the importance of the owner and other humans for subordinate and higher-ranking dogs when they face a difficult situation. According to our theory, the owner represents the most important resource for dogs [17,31,45,46]. Accordingly, dominant dogs have secured a more exclusive position for themselves with regard to the attention and interactions from their owner. In the present experiment, higher-ranking dogs more likely focused on individual problem solving in the non-social scenario, and they seemingly also tried to involve their owner by staring at him/her in the non-social test.

While the overall rank score did not have significant associations with behaviours in the two tests, its components (agonistic and leadership subscores) did. In the test where the dog had no active social partner, the agonistic score, an aspect of rank that is based completely on resource-related behaviours [32,35,47], affected not only dogs’ focus on the reward but also their determined, independent, persistent attempts to reach it. This is in line with previous findings on the association between personality traits and social rank where the more ‘conscientious’ dogs tended to rank higher. In the Canine Big Five, the trait ‘Conscientiousness’ includes items related to being focused, persistent, and goal-oriented [34]. Moreover, persistence is a key component of winning dominance contests and establishing higher rank in other species as well (e.g., green anole lizard [48]). The weaker determination (i.e., fewer cage-directed behaviours and staying closer to the owner) of dogs with lower agonistic scores in the non-social test can also be the consequence of their (original or learned) lower motivation in obtaining and possessing resources (see also [31]).

The other association we found in this test was between age and begging from the experimenter. The principal component ‘Begging’ included being close to and in physical contact with the experimenter, and we found that older dogs showed these behaviours less. An explanation for this could be that older dogs might be less pushy, or they are just less interested in the unfamiliar experimenter. Our results are in line with previous findings of an age-related decrease in stranger-directed contact behaviours (greeting and playfulness [49]) and social attention [50].There is also some evidence that ‘begging’ behaviour is more pronounced in lower-ranking and juvenile dogs, too. Cafazzo et al. [32] suggested that in subordinate dogs, the more frequent ‘begging’ displays could have their evolutionary origin in the food-requesting behaviour of juvenile individuals.

In the social test, the agonistic score had no effect, but the leadership score did. As this aspect of rank relies highly on social competence [39,51], it is not surprising that it had relevance in the social context only. On the other hand, it is worth noting that ‘social skills’ and ‘social competence’ when talking about rank mean intraspecific social competence, but the test had an interspecific social setting. It is important to remember that according to the principles of ethology, dogs and humans do not form rank-based hierarchies with each other [52]. Therefore, our results should not be interpreted to mean that dogs with a high leadership score would ‘use’ their possibly better social competence to outsmart or ‘dominate’ their human companions. On the contrary, when a previously rewarding person starts to behave in a frustrating way, dogs who usually collected positive experiences in social interactions may react with higher apparent stress. Dogs with higher leadership scores communicated more with the experimenter, although this communication showed some clear signs of frustration (e.g., barking can be a frustration-related vocalisation [17]). The leadership score in the DRA-Q consists of items related to being the initiating party in certain social situations (walking, facing threats) [31], and when these dogs were facing a difficult situation, their barking and prancing behaviours could be connected to their frustration of not being able to solve the problem in their usual proactive way.

The greatest limitation of this study was the exclusion of formal rank from the analysis. Formal displays are the most frequent rank-related behaviours in dog groups [53], so formal rank’s influence on behaviour in various contexts should be investigated. Unfortunately, as behaviours related to formal rank are less conspicuous than agonistic behaviours (e.g., winning fights) or group defence (e.g., barking at strangers), fewer owners can confidently respond to the questions regarding this aspect of rank. The exclusion of formal rank may lead to an overemphasis on overt dominance behaviours and situations of conflict (i.e., resource competition and group defence), oversimplifying the relationship between rank and behaviour.

Another limitation of this study is its low sample size, which made it impossible to investigate some possible confounding factors, such as training level and breed (as there were only one or two dogs from one breed), and to find possible interactions between variables. Future research on frustration should focus on these. Our test population was basically a convenience sample of dog breeds that also included mixed breeds. As we did not aim to find breed-related effects, a balanced convenience sample could be a suitable solution [54]. Although functional dog breeding could result in differences in the persistence and reliance on humans between cooperative and independent working dog breeds [13], and there are indications that genetic distance from their wolf-like ancestors can also affect persistence and human-directed gazing in dogs [55], we are not aware of any studies that discovered an association between dog breeds and rank position.

## 5. Conclusions

Our results not only show that social rank has an association with the frustration-related behaviours of family dogs but also that different aspects of rank correspond to different levels of human dependency and self-sufficiency. This is clearly noticeable in how agonistic rank and leadership have their own relevance in the contexts of different social natures and how dogs rather try to solve problems on their own or seek support from humans. These findings emphasise the complex nature of social rank and its influence on dogs’ behaviour. Instead of dominance as an umbrella term, it is more appropriate to use its separate elements when explaining dogs’ social behaviour based on the specific situation, when making broad generalisations across contexts based on either rank or even ‘subranks’ might be questionable.

As for the future, especially from the aspect of applied research, our results provide an interesting starting point for new experiments. It could be tested whether dominant or subordinate dogs would perform better in such scenarios, where the participation of a human handler is either unlikely, or, just the opposite, it is necessary. Based on the findings of this paper, we would predict that subordinate dogs would probably be better suited to such tasks, where they do not need to constantly rely on human guidance. However, dominant dogs would probably perform better when they need to work in close cooperation with their handler. Future research on the social behaviours and abilities of companion dogs should preferably also take into consideration the possible influence of the dog living in a single-dog versus multi-dog household, and the rank of the dog in the case of the latter, as it seems to not only affect intraspecific behaviours but behaviours in a broader range of social situations.

## Figures and Tables

**Figure 1 animals-14-03411-f001:**
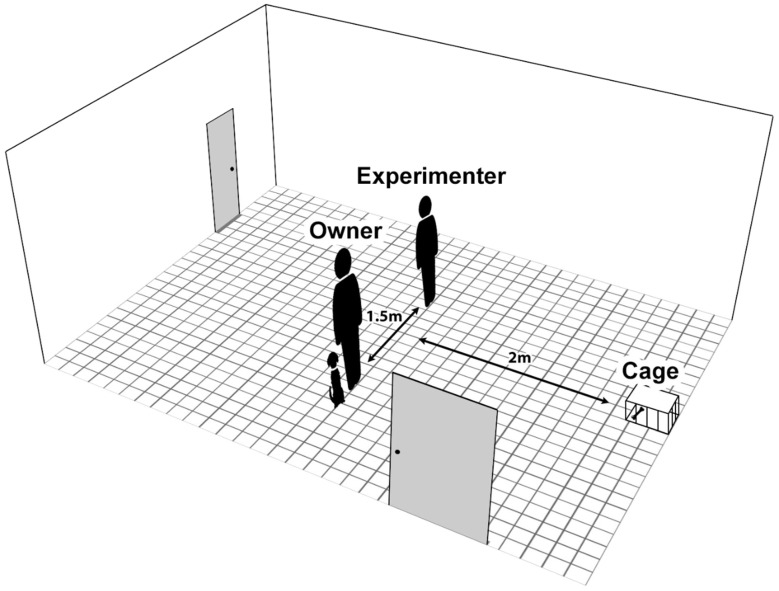
The experimental setup of the non-social frustration test.

**Figure 2 animals-14-03411-f002:**
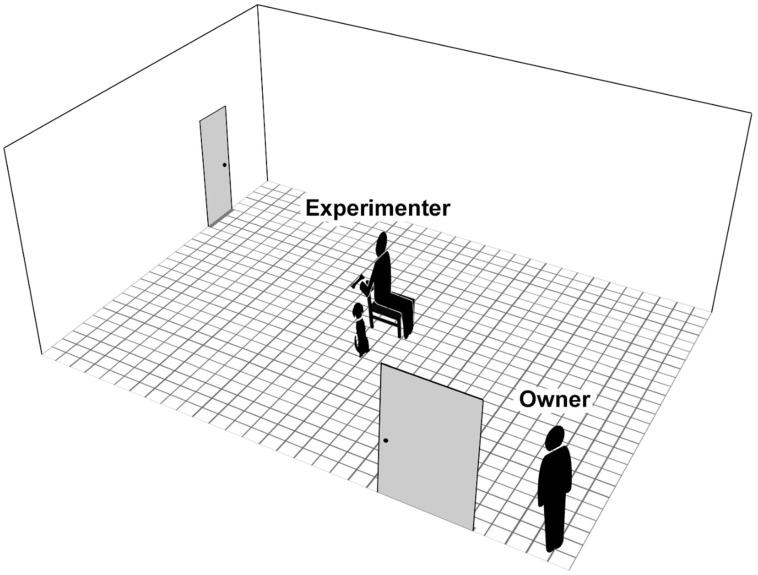
The experimental setup of the social frustration test.

**Figure 3 animals-14-03411-f003:**
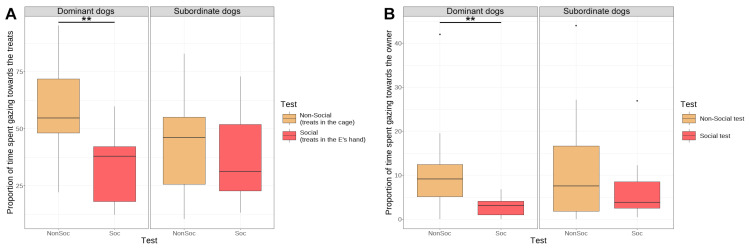
The results of the between-context (non-social vs. social test) comparisons for separate analyses of dominant and subordinate dogs in the case of (**A**) time spent gazing at the cage in the non-social test and the experimenter in the social test and (**B**) time spent gazing at the owner. The box shows the interquartile range between the lower and upper quartiles, divided by the median. The whiskers extend to the minimum and maximum values excluding the outliers; the dots show the outliers (more than 1.5 times the lower/upper quartile beyond the box). NonSoc = non-social test condition; Soc = social test condition; E = experimenter. The asterisks show significance in the following way: ** *p* <0.01.

**Figure 4 animals-14-03411-f004:**
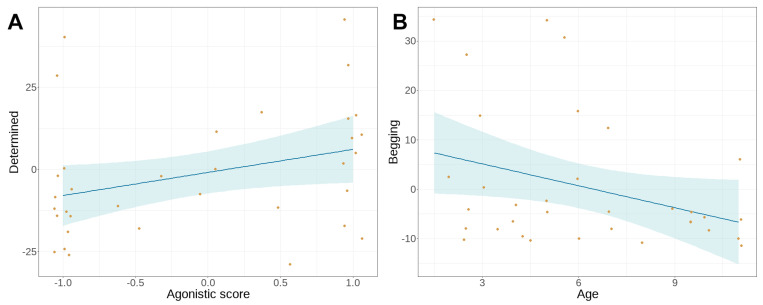
Significant associations in the non-social frustration test. (**A**) Dogs with higher agonistic scores were more determined to reach the treat; (**B**) older dogs showed less begging behaviour directed to the experimenter. The yellow dots are the datapoints, the blue band is the confidence interval.

**Figure 5 animals-14-03411-f005:**
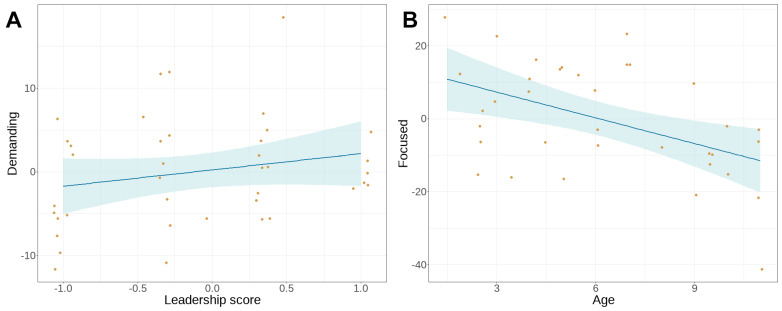
Significant associations in the social frustration test. (**A**) Dogs with higher leadership scores showed more demanding behaviour towards the experimenter; (**B**) older dogs were less focused on the experimenter and the treat. The yellow dots are the datapoints, the blue band is the confidence interval.

**Table 1 animals-14-03411-t001:** The variables coded in the social and non-social frustration tests.

Category	Name	Description	Variable Type
Position	Moving	The dog is moving, walking, or running, and 2–3 paws are on the ground the whole time	duration
	Sitting	The dog’s haunches are on the ground, but its elbows are not	duration
	Standing	The dog is standing on its 4 feet without locomotion	duration
	Lying	The dog’s elbows and sternum or side touch the ground	duration
Closeness	CageClose	The dog is within 50 cm of the cage *	duration
	OwnerClose	The dog is within 50 cm of the owner	duration
	ExperimenterClose	The dog is within 50 cm of the experimenter	duration
Gazing	CageGaze	The dog looks at the cage *	duration
	OwnerGaze	The dog looks at the owner	duration
	ExperimenterGaze	The dog looks at the experimenter	duration
Contact	CageContact	The dog is in physical contact with the cage/manipulating the cage (i.e., sniffing or poking with its nose, pawing or jumping on it) *	duration
	OwnerContact	The dog is in physical contact with the owner, poking with its nose, pawing or jumping on the owner	duration
	ExperimenterContact	The dog is in physical contact with the experimenter, poking with its nose, pawing or jumping on the experimenter	duration
Gaze alternation	OwnerGAlter	The dog quickly (within 2 s) alternates its gaze between the treat ** and the owner	frequency
	ExperimenterGAlter	The dog quickly alternates its gaze between the treat ** and the experimenter	frequency
Vocalisation	Barking	A loud, short, wide pitch range sound with inverted U-shaped pitch contour	duration
	Whining	High-pitched, relatively tonal, short and cyclic or elongated vocalisations	duration
	Other	Other types of vocalisations that do not fall into the other categories (growling, howling, moaning, coughing, sneezing, etc).	duration
	Panting	A noise made by the dog, which sounds like a loud, moderate to rapid, open-mouth respiration	duration
	Wagging	The dog is moving its tail constantly [not just because of the dogs’ movement (walking or running)]	duration
	Prancing in place	The dog quickly and repeatedly steps in one place, alternating its front legs	duration, frequency

* Cage-related behaviours were only coded in the non-social test. ** Cage in the non-social test, E’s hand in the social test.

**Table 2 animals-14-03411-t002:** Within-dog differences in behaviours between the non-social and social test in all dogs.

	All Dogs			
Behaviour	W	Estimate	*p* *	r
**Moving**	661	20.1	**<0.0001**	0.683
**Sitting**	5	−29.9	**<0.0001**	0.856
**Standing**	619	17.2	**0.0004**	0.585
*Lying*	55	−14.3	*0.0764*	0.349
**CageClose/ExperimenterClose**	82.5	−29.4	**<0.0001**	0.678
CageContact/ExperimenterContact	415	1.89	0.3705	0.153
**CageGaze/ExperimenterGaze**	572	16.9	**0.0051**	0.474
**OwnerGaze**	596	5.61	**0.0024**	0.531
**Wagging**	498	14.5	**0.0051**	0.475
**Panting**	150	−9.78	**0.0054**	0.459
**ExpGazeAlter**	0	4.5	**<0.0001**	0.794
Vocalisation	112	−0.5	0.6490	0.192
**Prancing**	5.5	−2.25	**0.0051**	0.522

Significant differences are in bold; non-significant trends are in italic. ExpGazeAlter = gaze alternation between the experimenter and the cage (non-social test) or the experimenter’s hand (social test). * adjusted *p* values.

**Table 3 animals-14-03411-t003:** Within-dog behavioural differences between the social and non-social contexts, separately in dominant and subordinate dogs.

	Dominant Dogs				Subordinate Dogs			
Behaviour	W	Estimate	*p* *	r	W	Estimate	*p* *	r
**Moving**	144	20.6	**0.0033**	0.775	132	26.2	**0.0144**	0.637
**Sitting**	0	−38.1	**0.0003**	0.873	3	−21.4	**0.0010**	0.844
Standing	113	12.1	0.1280	0.419	134	21.4	**0.0144**	0.660
Lying	19	−7	0.4150	0.353	8	−35.7	*0.0926*	0.401
**CageClose/ExperimenterClose**	30	−14.3	**0.0478**	0.534	5	−41.9	**0.0010**	0.821
CageContact/ExperimenterContact	96	4.1	0.1832	0.345	93	1.39	0.4864	0.190
CageGaze/ExperimenterGaze	141	24.8	**0.0045**	0.741	94.5	7.22	0.4810	0.207
OwnerGaze	144	6.28	**0.0045**	0.781	107	4.83	0.2028	0.350
Wagging	76	11.6	0.1832	0.340	117	11.5	*0.0925*	0.465
**Panting**	18	−13.7	**0.0399**	0.586	20	−11.1	**0.0144**	0.649
**ExpGazeAlter**	0	−5.5	**0.0099**	0.778	0	−3.5	**0.0067**	0.830
Vocalisation	19	−1.06	0.4150	0.353	24	−0.28	0.7600	0.126
Prancing	0	−2.5	**0.0425**	0.635	3	−1.5	0.2028	0.400

Significant differences between the two tests are in bold; non-significant trends are in italic. Behaviours that did not change similarly in the two groups are underlined. * adjusted *p* values.

**Table 4 animals-14-03411-t004:** The principal components obtained from the PCA on behaviours in each test.

Non-Social Test					
	Determined	Hopeful	Complaining	Passive	Begging
CageGaze	0.8433				
CageContact	0.8021				
CageClose	0.7279				
OwnerClose	−0.6188				
ExperimenterGaze		0.7318			
Lay		0.6665			
Move		−0.7952			
Vocal			0.9625		
Whining			0.9597		
Sit				0.5170	
Stand				−0.9587	
ExperimenterClose					0.8814
ExperimenterContact					0.8388
Cronbach’s α	0.778	0.653	0.935	0.529	0.714
**Social Test**					
	**Demanding**	**Complaining**	**Focused**	**Passive**	**Excited**
Barking	0.8856				
Other	0.8812				
Prancing	0.6248				
Whining		0.7885			
Lay		0.7463			
OwnerGaze		0.7415			
Sit		−0.5593			
ExperimenterClose			0.6224		
OwnerClose			−0.6287		
Stand			−0.6468		
OwnerContact			−0.6701		
ExperimenterGaze				0.7735	
ExperimenterContact				−0.5682	
Move				−0.7954	
Panting					0.6967
Wagging					0.6663
Cronbach’s α	0.794	0.698	0.609	0.553	0.281

## Data Availability

The dataset used for the analyses is provided in the Appendix A.

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
