# Peer review of "Rank-Related Differences in Dogs’ Behaviours in Frustrating Situations"

_animals, 2024, doi:10.3390/ani14233411_

Round 1
Reviewer 1 Report
Comments and Suggestions for Authors
Thank you for the opportunity to review this article. Research on dog behaviour is always interesting and I commend the authors for their work in this area.
Unfortunately, I cannot recommend this article for publication. My main criticism is that the interpretation of the results do not appear to match the reported results. The authors claim there are differences in behaviour between dominant and subordinate dogs (Tables 2 and 3 and Figure 3) but I disagree that these differences exist. Rather, it appears to be a problem of statistical power because all differences are in the same direction but do not always reach statistical significance. Other comments are described below that may be helpful to the authors.
The Introduction is difficult to understand as the writing is very confusing in places. As a behaviour analyst, I had trouble with the many mentalistic explanations of behaviour, which I describe in more detail below. Related to this point, given that the paper was about “frustration”, there was no attempt to explain why certain behaviours were indicative of frustration. The researchers measured observable behaviours, such as sitting, moving, staring, etc, but did not explain how these behaviours indicated that the dogs were “frustrated”.
On Lines 69-70, and elsewhere in the manuscript, the authors state that frustration can arise from the failure to deliver an “expected” reward. As a behaviour analyst, I would prefer non-mentalistic and non-anthropomorphic language, such as “behaviours consistent with frustration can arise from the withholding of rewards for behaviours that have been previously reinforced.” We cannot know what the dog “expected”, we can only report on observable events, which is that reinforcers were not delivered.
Line 76: what does it mean for a dog to “have control” over the situation? I would recommend avoiding this word as it simply raises more questions. The word “access” seems to have the intended meaning without creating a need to define what is meant by “control”.
Line 80: Sentence starts “This was especially typical in dogs…” It is not clear what “this” refers to.
Line 93: A questionnaire cannot show canine traits – it can only reflect what dog owners report those traits to be. A measure of “traits” (or more accurately behaviours) can only be undertaken by direct observation of the dog’s behaviour.
Line 102-103: I also disagree here that “personality traits” can cause behaviour. Personality traits are category labels for types of behaviours demonstrated by dogs. If the rank of a dog is associated with personality traits, then it is the rank that causes both the personality traits and the behaviour.
Line 113: You mention here that a “third aspect” is leadership. It is not clear to me what the other two are, or even what you are referring to. Aspect of what? Given that you go on to examine the data according to leadership and agonistic scores, you need to much more clearly define what you are talking about. I went to the authors’ previously published article where they developed the DRA-Q dominance scale (Vékony et al., 2022) and could see no mention of subscales relating to agonistic behaviour or leadership. Which of the questionnaire items contribute to these subscale scores?
I assume the study was not pre-registered? Either way, the results of the study should be presented separately for hypothesized and exploratory results. It seems that most of the results relate to effects that were not hypothesized. These results should be interpreted more cautiously than hypothesized effects as the study was not designed to test these effects.
Hypothesis: The first hypothesis has two parts: 1) that in the asocial context (note, please use consistent terms, elsewhere this condition is referred to as the non-social context) there would be no effect of rank, and 2) that agonistic dogs “might” be more focused on the reward. It is not good practice to hypothesise a null effect (as in 1). And, in (2), were they expected to be focused on the reward or not? Stating that they “might” seems like a weak prediction. Overall, the aims and hypotheses could be expressed much more clearly.
How many dogs were allocated to the dominant and subordinate groups? Although this information is in the supplementary tables, it should be reported in the main text. Relatedly, what were the degrees of freedom for the various statistical analyses? This is important for interpreting the statistical results as data from small samples are likely to be more variable. Differences in the variability of the data can be seen in Figure 3. More variability in the data for could also contribute to the non-significant effects such as those reported in Table 3. Additionally, sample sizes for the dominant and subordinate groups are both the recommended minimum size for enable us to assume that the sampling distribution of the mean is normal. The authors have not commented on whether their data meet the assumptions of the statistical tests that they have performed.
Note that all statistical tests should also be accompanied by effect sizes.
Line 250-252: you state that the results in Table 2 for all dogs together were not “universal” (please choose a better word here as it is not clear what you mean, but I will assume you mean that the results are not the same) in Table 3 when the dogs were separated by rank. What patterns of behaviour were not the same? Looking at the two tables, all the significant differences between non-social and social data in Table 2 were in the same direction in Table 3 for both dominant and subordinate dogs. Does this not indicate that the patterns of behaviour were rather similar whether the dogs were grouped or separated by rank? As stated above, the fact that some significant effects were no longer significant when the data were split by rank does not indicate that the data were different as this could be an issue with statistical power. In Figure 3, for example, three behaviours have been singled out as being different across dominant and subordinate dogs. Figure 3 clearly shows that the data were more variable for one group compared to the other (particularly for Panels A and C), but in every case, the means are in the same direction for both groups of dogs. I do therefore not see the difference the authors purport. Effect sizes would help to support (or not) the claims about differential behaviour of the groups.
In the Discussion (Lines 321-326) the authors state that dominant and subordinate dogs behaved differently towards the experimenter and the cage when the reward became inaccessible. As mentioned above, I don’t think the authors can claim this. The statements of results in this part of the Discussion also do not seem to align with the claims in the Abstract (“”…the holistic rank (‘dominant’ vs. ‘subordinate’) of the dogs did not show significant association with their reactions to frustrating situations.”)
Line 336: what does it mean for a dog to show a “marked ignorance” towards the experimenter? What was the dog meant to “know” about the experimenter?
Line 342: “The weaker persistence of subordinate dogs in the non-social test…” how was this “weaker persistence” demonstrated in the data? By which measures?
Comments on the Quality of English LanguageThe writing is confusing in places and could be improved
Author Response
RESPONSE TO REVIEWER 1
Thank you for the opportunity to review this article. Research on dog behaviour is always interesting and I commend the authors for their work in this area.
Unfortunately, I cannot recommend this article for publication. My main criticism is that the interpretation of the results do not appear to match the reported results. The authors claim there are differences in behaviour between dominant and subordinate dogs (Tables 2 and 3 and Figure 3) but I disagree that these differences exist. Rather, it appears to be a problem of statistical power because all differences are in the same direction but do not always reach statistical significance. Other comments are described below that may be helpful to the authors.
RESPONSE: Thank you for the detailed evaluation of our paper. We have realized that the original text could be clearer at particular sections of the manuscript, and this could cause misunderstandings. We did our best to clarify the problematic points either in the manuscript by rewriting some sections, or here, in the Revision Note.
Regarding the statistical analysis, we had two main, separate approaches. First, with paired t-tests, we compared at first the behavior of each dog (regardless to their rank) between the two contexts (non-social vs. social) – Table 2. Also, with paired t-tests, we compared dogs’ behavior between the two contexts, but at this time separately in the subordinate and dominant dog groups (Table 3). Therefore, these analyses did not compare dominant dogs to subordinate ones, but based on the results that are shown in Table 3, we could show whether the differences in the dogs’ responses caused by the social/non-social nature of the task were different in the case of the dominant and subordinate dogs (Figure 3). In the legend of Figure 3 we highlighted that the corresponding statistical analysis has been run separately for the dominant and subordinate dogs.
The second round of statistical analyses contained those tests where we added rank scores (and subranks) as fixed factors to the models, and directly analyzed their effect on the dogs’ behavior. In these tests we used derived behavioral variables, resulting from the Principal Component Analysis.
The Introduction is difficult to understand as the writing is very confusing in places.
RESPONSE: Now we used a native English speaker colleague to help us proofreading the text. We hope that the Introduction became clearer and has a better flow.
As a behaviour analyst, I had trouble with the many mentalistic explanations of behaviour, which I describe in more detail below.
RESPONSE: We did our best to clarify these points, see below.
Related to this point, given that the paper was about “frustration”, there was no attempt to explain why certain behaviours were indicative of frustration. The researchers measured observable behaviours, such as sitting, moving, staring, etc, but did not explain how these behaviours indicated that the dogs were “frustrated”.
RESPONSE: Thank you for providing us the opportunity to clarify some aspects of our research. As the article’s title states, we did not directly analyze dogs’ ‘frustration’, but we compared their reactions between social and non-social situations that are considered as being ‘frustrating’ (i.e., when a formerly accessible reward is being denied). However, we agree with the Reviewer that it would be useful to show whether the behaviors we observed in our tests show resemblance with the behaviors that other researchers found in their frustration-related research. There are two papers that can be related to our study. Jakovcevic et al. (2013) used a very similar social context to the one we applied here: dogs were at first rewarded for establishing eye-contact with the experimenter, then the reward was stopped to be provided. Here, the authors described the following behavioral variables as indicators of canine frustration: “The dog's frustration reactions during extinction involved an increase in withdrawal and side orientation to the location of the human as well as lying down, ambulation, sniffing, and vocalizations compared with the last acquisition trial.” From these, vocalizations, sniffing (‘contact’, ‘experimenter close’), withdrawal, lying and ambulation (‘moving’) can be found in our complex behavioral components as well (‘Demanding’, ‘Complain’, ‘Focused’, ‘Passive’). The other research we can refer is the one from Lenkei et al. (2021), where dogs were tested in a short indoor separation context. Here, in connection with a questionnaire that was completed by the dogs’ owners, frustration-prone dogs scratched the door more, and barked more frequently than dogs that were less prone to frustration. In our present study, barking was part of the ‘Demanding’ behavioral component in the social context, and dogs tried to get the reward from the cage by physically contacting the cage in the non-social context as well (‘Determined’ component). References:
Jakovcevic, A., Elgier, A. M., Mustaca, A. E., & Bentosela, M. (2013). Frustration behaviors in domestic dogs. Journal of applied animal welfare science, 16(1), 19-34.
Lenkei, R., Faragó, T., Bakos, V., & Pongrácz, P. (2021). Separation-related behavior of dogs shows association with their reactions to everyday situations that may elicit frustration or fear. Scientific Reports, 11(1), 19207.
On Lines 69-70, and elsewhere in the manuscript, the authors state that frustration can arise from the failure to deliver an “expected” reward. As a behaviour analyst, I would prefer non-mentalistic and non-anthropomorphic language, such as “behaviours consistent with frustration can arise from the withholding of rewards for behaviours that have been previously reinforced.” We cannot know what the dog “expected”, we can only report on observable events, which is that reinforcers were not delivered.
RESPONSE: The Reviewer’s comment was somewhat surprising as the term “expectation/ expecting” has a well-established usage in behavioral literature (for example, even in the case of insects: Gil, M., & De Marco, R. J. (2009). Honeybees learn the sign and magnitude of reward variations. Journal of Experimental Biology, 212(17), 2830-2834.). Biological definitions of frustration also rely on the term “expectation”, even in non-human animals (e.g., Burokas, A., Gutiérrez‐Cuesta, J., Martín‐García, E., & Maldonado, R. (2012). Operant model of frustrated expected reward in mice. Addiction biology, 17(4), 770-782.). There is a whole phenomenon with a long list of publications, called ‘violation of expectation’ paradigm (see for a review, Margoni, F., Surian, L., & Baillargeon, R. (2024). The violation-of-expectation paradigm: A conceptual overview. Psychological Review, 131(3), 716.).
However, we felt that our manuscript would be also understandable for the reader without saying that the dogs ‘expected’ something – therefore we omitted this term with regard to the dogs, and kept it only when it is connected to the definition of frustration, or to the authors’ expectation about particular results.
Line 76: what does it mean for a dog to “have control” over the situation? I would recommend avoiding this word as it simply raises more questions. The word “access” seems to have the intended meaning without creating a need to define what is meant by “control”.
RESPONSE: Thank you for the comment. We rewrote this section, avoiding the term “control” in connection to dogs (lines 75-80). The new section reads like this:
“Dogs’ reaction to a delayed, blocked or absent positive stimulus (or resource) can depend on their previous experiences with the situation (the more they know how to operate an unanimated food-dispenser device successfully, the less stress signs they show when the device stops providing the reward [23]). When dogs are separated from their caretaker (i.e., they are left at home alone), they lose access to their single most crucial resource, and this is thought to be a main contributor to separation-related problems.”
As a side note, we thought that the terminology “having control” over particular effects of the environment has been well-established in animal behavioral research (for example, Weiss, J. M. (1971). Effects of coping behavior in different warning signal conditions on stress pathology in rats. Journal of Comparative and Physiological Psychology, 77(1), 1.)
Line 76: Sentence starts “This was especially typical in dogs…” It is not clear what “this” refers to.
RESPONSE: Thank you for calling our attention to the potentially confusing wording here. We rewrote this section (lines 78-83), now it reads like this:
“When dogs are separated from their caretaker (i.e., they are left at home alone), they lose access to their single most crucial resource, and this is thought to be a main contributor to separation-related problems. It was found that those dogs reacted with frequent barking to a short separation episode mostly that lived with ‘lenient owners’, who normally easily give in when the dog behaves demandingly [24].”
Line 89: A questionnaire cannot show canine traits – it can only reflect what dog owners report those traits to be. A measure of “traits” (or more accurately behaviours) can only be undertaken by direct observation of the dog’s behaviour.
RESPONSE: Thank you for this cautionary comment. Actually, in our manuscript we did not only rely to a paper that wrote about the Canine Frustration Questionnaire, but cited a further paper from those authors (McPeake, K. J., Collins, L. M., Zulch, H., & Mills, D. S. (2021). Behavioural and physiological correlates of the Canine Frustration Questionnaire. Animals, 11(12), 3346.), where they validated the questionnaire with behavioral test and also with cortisol assay. Therefore, we think that in this case the original text can be kept.
If questionnaires are properly validated, they represent a widely useful, reliable toolkit for assessing various traits of animals (e.g., animal personalities, Ley, J. M., Bennett, P. C., & Coleman, G. J. (2009). A refinement and validation of the Monash Canine Personality Questionnaire (MCPQ). Applied Animal Behaviour Science, 116(2-4), 220-227. Tetley, C. L., & O’Hara, S. J. (2012). Ratings of animal personality as a tool for improving the breeding, management and welfare of zoo mammals. Animal Welfare, 21(4), 463-476.)
Line 96: I also disagree here that “personality traits” can cause behaviour. Personality traits are category labels for types of behaviours demonstrated by dogs. If the rank of a dog is associated with personality traits, then it is the rank that causes both the personality traits and the behaviour.
RESPONSE: We totally agree with the Reviewer, and in the original text we were cautious enough to avoid such claims that frustration-related responses or dogs’ rank would be the consequence of their personality traits. However, we added the term “non-causative” to this section, and we hope that now it reads even clearer (lines 102-111):
“Position of an individual within the hierarchy develops through social interactions [31] and may even change dynamically during the life of the dog [32]. Following the well-established ethological definitions [33], a ‘dominant dog’ is the individual who regularly prevails over the other cohabitant or group member dog (the ‘subordinate’) in dyadic competitive events, and as consequence, gains priority to resources. There are indications that among dogs, a well-established hierarchy is mostly maintained by signals of submission from the subordinate dogs rather than by repeated displays or acts of aggression [32]. Personality traits (e.g., extraversion, openness, conscientiousness, agreeableness) that show a non-causative association with the rank-position of an individual dog [34], may also be influential in their reactions to frustration-generating situations.”
Line 113: You mention here that a “third aspect” is leadership. It is not clear to me what the other two are, or even what you are referring to. Aspect of what?
RESPONSE: We agree that in this section, the three separate aspects of the ‘holistic’ rank (Formal, Agonistic and Leadership) were not clearly enough highlighted. We rewrote the section and also used typographic methods to highlight the three aspects of rank now (lines 124-146).
Given that you go on to examine the data according to leadership and agonistic scores, you need to much more clearly define what you are talking about. I went to the authors’ previously published article where they developed the DRA-Q dominance scale (Vékony et al., 2022) and could see no mention of subscales relating to agonistic behaviour or leadership. Which of the questionnaire items contribute to these subscale scores?
RESPONSE: We added clearer descriptions for the three aspects of dominance rank. The reference correctly for our recent paper on this topic is:
The section reads now like this (lines 124-146):
“While the position of an individual in the hierarchy can be mainly tracked back to resource-related interactions, social rank has other aspects and presentations that are not strictly tied to resource competition. In one recent paper, a complex approach was used to social rank, which included three distinct aspects [31]. The association of resource-possession and greeting behaviour with Formal, Agonistic, and Leadership-rank scores in cohabiting dogs was investigated by Vékony and Pongrácz [31], and the results indicated that the various rank components’ effects may depend, to a different extent, on the dog’s social skills. This suggests that they may also have different relevance in social and non-social situations. Formal rank can be described as directionally consistent affiliative displays between dominant and subordinate individuals [38]. While originally thought to be a direct extension of the established Agonistic rank – this rank is determined by competitive behaviours. Research on free-ranging dogs found that not every individual consistently receives submissive displays in agonistic contexts from the others, but also receives formal submission (outside of direct competition). Usually, younger individuals with high agonistic rank do not possess the same formal rank, suggesting that formal rank requires social skills more than competitive successes [39]. A third aspect, related to, but does not necessarily coincide completely with the previous two, is ‘Leadership’, which was found to be related more to affiliative relationships and formal rank in the case of group movement leadership [39], but only to some extent and in certain cases of leading territorial defence [40]. Although the companion dogs’ environment is much more relaxed in terms of need for resource competition or territorial defence, and much more controlled (by humans) in terms of choosing group-mates and almost all aspects of life, agonistic and formal-affiliative displays [31,41] and the leading of group movements [42] were also described in both temporary and cohabiting dog groups.”
I assume the study was not pre-registered? Either way, the results of the study should be presented separately for hypothesized and exploratory results. It seems that most of the results relate to effects that were not hypothesized. These results should be interpreted more cautiously than hypothesized effects as the study was not designed to test these effects.
RESPONSE: This was not a pre-registered study. We had a priori hypotheses and predictions for most of the comparisons we report then in the Results chapter. The only exception is probably the analysis we report in Table 2, where we compared behaviours in the two tests on the whole sample, independently of dogs’ rank. With this analysis, we showed the general effect of the social/non-social context. We report this analysis separately from the others (which we had corresponding hypothesis/prediction), we think that we fulfilled the request of the Reviewer.
Hypothesis: The first hypothesis has two parts: 1) that in the asocial context (note, please use consistent terms, elsewhere this condition is referred to as the non-social context) there would be no effect of rank, and 2) that agonistic dogs “might” be more focused on the reward. It is not good practice to hypothesise a null effect (as in 1). And, in (2), were they expected to be focused on the reward or not? Stating that they “might” seems like a weak prediction. Overall, the aims and hypotheses could be expressed much more clearly.
RESPONSE: We checked the manuscript for inconsistencies in the terminology, and now we use only “non-social” in the case of the context where the reward was placed into the cage. We rewrote the aims/hypothesis/predictions paragraphs, aiming to a more straightforward and understandable structure (lines 148-174). We named two main aims: (1) to examine the effect of social rank on the responses of dogs to frustration-generating situations; and (2) to detect the potential effect of the social or non-social nature of the context on the rank-dependent reactions of the dogs. With regard to the ‘null-effect’, this is in one of our predictions, where “We predicted that in the non-social context overall rank would not affect dogs’ persistence to obtain the reward or target-directed behaviours, but more agonistic dogs might be more focused on the reward, because their rank was affected stronger by their capacity to win resource-related contests.”
How many dogs were allocated to the dominant and subordinate groups? Although this information is in the supplementary tables, it should be reported in the main text.
RESPONSE: We had 18 subordinate, 17 dominant and 2 flexible rank dogs in the sample (meaning: two cohabiting dogs received the exact same score). There were also two dogs from 3-dog households that ranked between the other two dogs, thus could not be described with this binary approach.
Relatedly, what were the degrees of freedom for the various statistical analyses? This is important for interpreting the statistical results as data from small samples are likely to be more variable. Differences in the variability of the data can be seen in Figure 3. More variability in the data for could also contribute to the non-significant effects such as those reported in Table 3. Additionally, sample sizes for the dominant and subordinate groups are both the recommended minimum size for enable us to assume that the sampling distribution of the mean is normal. The authors have not commented on whether their data meet the assumptions of the statistical tests that they have performed.
Note that all statistical tests should also be accompanied by effect sizes.
RESPONSE: We are grateful to the Reviewer for this comment. Originally, we have checked the distribution of data only before performing the models in the case of the principal components. This truly was not the best practice, and now, complying with the Reviewer’s recommendation, we also checked our data distributions in the case of the less complex statistical analyses. Consequently, we changed our methods to non-parametric statistical tests, and instead of paired t-tests, now we run Wilcoxon signed-rank tests. According to the new statistics, almost all the previously reported significant associations remained, with convincing effect sizes. Thank you once more the advice.
Line 250-252: you state that the results in Table 2 for all dogs together were not “universal” (please choose a better word here as it is not clear what you mean, but I will assume you mean that the results are not the same) in Table 3 when the dogs were separated by rank.
RESPONSE: We rewrote this sentence, now it reads like this (lines 283-285):
“The analysis of the effect of test context in dominant and subordinate dogs separately revealed that the results in the case of all dogs cannot be generalized to the dominant and subordinate dogs (Table 3).”
What patterns of behaviour were not the same? Looking at the two tables, all the significant differences between non-social and social data in Table 2 were in the same direction in Table 3 for both dominant and subordinate dogs. Does this not indicate that the patterns of behaviour were rather similar whether the dogs were grouped or separated by rank?
RESPONSE: Looking at Tables 2 and 3 it is well-visible that when dominant and subordinate dogs were separately analyzed, not always the same behaviors showed significant effect of the context. Behaviors that were only significant in the case of high-ranking dogs: CageGaze/ ExperimenterGaze, OwnerGaze, Prancing. Behaviour that was only significant in the case of low-ranking dogs: Standing. Behaviours showing a trend only in the case of low-ranking dogs are Lying and Wagging.
As stated above, the fact that some significant effects were no longer significant when the data were split by rank does not indicate that the data were different as this could be an issue with statistical power.
RESPONSE: From the 10 behaviors that showed a significant association with context in the case of the analysis on the full sample (Table 2), only one (Wagging) did not show any significant effect in the case of separately analyzed dominant and subordinate dogs (Table 3). The other 8 variables showed significant association in one or both groups.
In Figure 3, for example, three behaviours have been singled out as being different across dominant and subordinate dogs. Figure 3 clearly shows that the data were more variable for one group compared to the other (particularly for Panels A and C), but in every case, the means are in the same direction for both groups of dogs. I do therefore not see the difference the authors purport. Effect sizes would help to support (or not) the claims about differential behaviour of the groups.
RESPONSE: Figure 3 was redrawn according to the new (Wilcoxon) statistics. We removed the panel that earlier showed the results of ‘CageClose/ExperimenterClose’ results, because according the new Wilcoxon-tests, the effect of context was the same in subordinate and dominant dogs.
In the Discussion (Lines 321-326) the authors state that dominant and subordinate dogs behaved differently towards the experimenter and the cage when the reward became inaccessible. As mentioned above, I don’t think the authors can claim this. The statements of results in this part of the Discussion also do not seem to align with the claims in the Abstract (“”…the holistic rank (‘dominant’ vs. ‘subordinate’) of the dogs did not show significant association with their reactions to frustrating situations.”)
RESPONSE: We partially rewrote this paragraph, as with the newly run Wilcoxon tests the results became somewhat different. Now, both dominant and subordinate dogs acted differently in the social and non-social test contexts (regarding closeness to the cage/experimenter). However, we would like to highlight that in this paragraph, we discuss those results that are not about the direct comparison among the subjects based on their rank scores. With the Wilcoxon tests (just as before with the paired t-tests) we analyzed the effect of test context (social vs. non-social) on either the whole population of subjects, or separately within the dominant and subordinate dogs.
Line 336: what does it mean for a dog to show a “marked ignorance” towards the experimenter? What was the dog meant to “know” about the experimenter?
RESPONSE: Based on the new statistics, we rewrote this paragraph, and deleted the “marked ignorance” phrase, too.
Line 342: “The weaker persistence of subordinate dogs in the non-social test…” how was this “weaker persistence” demonstrated in the data? By which measures?
RESPONSE: Thank you for this comment, “persistence” was not the best term in this case. We rewrote this section (lines 380-383), and now we refer to the correct name of the principal component (‘determined’). We also indicated the involved behaviors. The text reads like this:
“The weaker determination (i.e., less cage-directed behaviours and staying closer to the owner) of dogs with lower Agonistic scores in the non-social test, can also be the consequence of their (original or learned) lower motivation in obtaining and possessing resources (see also [31]).”
Reviewer 2 Report
Comments and Suggestions for Authors
General comments
The article investigates rank-related behavioral differences in dogs' reactions to frustrating situations. It uses a solid methodology, including validated questionnaires and well-structured frustration tests. The study's main strengths are its clear research hypothesis and the novel approach of investigating different aspects of social rank (Agonistic, Leadership) rather than relying solely on the holistic rank score. However, there are some minor issues related to the clarity of certain statistical results and the exclusion of Formal rank from the analysis, which could limit the study’s interpretability
Detailed comments
Line 13-22: the introduction effectively sets up the rationale for the study. However, it would benefit from more explicit connections between social rank in cohabiting dogs and the study's aims. Consider expanding this section to clarify how past studies inform the current research focus.
Line 150-160: the exclusion of one subject due to owner non-compliance is noted, but further clarification could be added regarding how this impacted the study's results, especially given the small sample size (N = 39 dogs). Does this exclusion introduce any biases?
Line 163-185: the setup and procedures are described clearly. However, it would be helpful to include more details on how the dog’s behaviors were coded in real time and if there was any inter-rater reliability check among the people who coded.
Line 193-203: the two-minute duration in the training phase seems relatively short for dogs to establish a solid understanding of the task. Could this have impacted the results, especially for more cautious or slower-learning dogs?
Line 233-249: the description of how the five principal components were derived could benefit from further explanation. Specifically, how were the cutoff points for inclusion in each principal component determined?
Line 290-299: concerning decreased begging behavior, is this finding consistent with other studies involving canine frustration or cognitive decline with age?
Line 304-307: the association between agonistic scores and determined behavior is very interesting, but did the authors find any confounding factors, such as breed type or prior training. Was it tested?
Line 319-330: the results showing differences in experimenter-directed behaviors between dominant and subordinate dogs could be strengthened by further qualitative descriptions of what these behaviors looked like in practice.
Line 371-376: the authors acknowledge the exclusion of formal rank from the analysis, but this limitation should be more thoroughly addressed, particularly regarding how it might have influenced the findings.
Author Response
RESPONSES TO REVIEWER 2
General comments
The article investigates rank-related behavioral differences in dogs' reactions to frustrating situations. It uses a solid methodology, including validated questionnaires and well-structured frustration tests. The study's main strengths are its clear research hypothesis and the novel approach of investigating different aspects of social rank (Agonistic, Leadership) rather than relying solely on the holistic rank score.
RESPONSE: We are grateful for the Reviewer’s supportive attitude and for the useful suggestions, comments. We provide point-by-point answers to these in the next section, and where it was necessary, we changed the manuscript accordingly.
However, there are some minor issues related to the clarity of certain statistical results and the exclusion of Formal rank from the analysis, which could limit the study’s interpretability
Detailed comments
Line 13-22: the introduction effectively sets up the rationale for the study. However, it would benefit from more explicit connections between social rank in cohabiting dogs and the study's aims. Consider expanding this section to clarify how past studies inform the current research focus.
RESPONSE: Thank you for this suggestion. We restructured the aims and hypotheses/predictions section of the Introduction, and following the Reviewer’s request, we also added this text, with more details on the possible connection between previous research on dogs’ hierarchy and the present study’s goals (Lines 112-123).
“In the case of dogs, the strong social bonds with humans, and especially with the owner, is of paramount importance. According to one theory, the owner can be the most important and hard-to-be-shared resource for cohabiting companion dogs [31], and because of this, the dogs will show rank-related behavioural differences in such contexts that involve interactions with the owner, or to a broader extent, with humans. There are earlier indications that higher-ranking dogs rely more on human behaviour during observational learning tasks than subordinate dogs do [35,36], which may suggest that when dogs face a frustrating situation, dominant individuals would also more likely turn towards nearby humans for assistance. Higher rank in dogs has been found to be connected to behaviours such as possession [31,37], dog-rivalry/ aggression, impulsivity [37], which allows the assumption that dominant dogs would show more intense signs of frustration when their demanding needs are not fulfilled.”
Line 150-160: the exclusion of one subject due to owner non-compliance is noted, but further clarification could be added regarding how this impacted the study's results, especially given the small sample size (N = 39 dogs). Does this exclusion introduce any biases?
RESPONSE: In the case of the excluded dog, unfortunately the owner intervened during the test, by physically pushing the subject towards the cage (non-social context). Therefore, we did not evaluate the behavior of this dog, as this would cause the bias in our data set.
Line 163-185: the setup and procedures are described clearly. However, it would be helpful to include more details on how the dog’s behaviors were coded in real time and if there was any inter-rater reliability check among the people who coded.
RESPONSE: The behavioral coding is described in section 2.4 Behavioural coding (Line 219). All coding was done from video recordings of the camera and microphone system. Thank you for requesting the inter-rater reliability details, because this important step somehow has been forgotten to be added to the original manuscript. Now it is included to the Methods and Results sections (Lines 224, 250-253).
Line 193-203: the two-minute duration in the training phase seems relatively short for dogs to establish a solid understanding of the task. Could this have impacted the results, especially for more cautious or slower-learning dogs?
RESPONSE: The 2-min long training phase was chosen based on earlier publications (e.g., Bentosela, M., Barrera, G., Jakovcevic, A., Elgier, A. M., & Mustaca, A. E. (2008). Effect of reinforcement, reinforcer omission and extinction on a communicative response in domestic dogs (Canis familiaris). Behavioural processes, 78(3), 464-469.), and also on our own experience with the typical subjects we had chance to encounter lately. The companion dogs in our sample were all well-socialized and food-motivated subjects that readily engaged eye-contact with the experimenter during the training phase. Importantly, our method did not require that the dogs would establish longer and longer eye contact with the experimenter, or do this behavior with a decreasing latency.
Line 233-249: the description of how the five principal components were derived could benefit from further explanation. Specifically, how were the cutoff points for inclusion in each principal component determined?
RESPONSE: A usual cutoff point for PCA in the literature is 0.4, but these are most often questionnaire studies (e.g., Canine Frustration Questionnaire by McPeake et al. 2019, translation of the MDORS by Houtert et al., 2019 or Parenting Styles by Herwijnen et al., 2018) with very large samples sizes, and the empirical ones also usually operate with larger samples than what we had (e.g., VIDEOPET study with 217 border collies by Turcsán et al., 2018 or detour study with 98 dogs by Pongrácz et al., 2021). By opting for a stricter, 0.5 cutoff, we tried to avoid overinterpreting weaker loadings, which is a risk in case of smaller sample sizes.
McPeake, K. J., Collins, L. M., Zulch, H., & Mills, D. S. (2019). The Canine Frustration Questionnaire—Development of a New Psychometric Tool for Measuring Frustration in Domestic Dogs (Canis familiaris). Frontiers in Veterinary Science, 6, 454627. https://doi.org/10.3389/fvets.2019.00152
Van Houtert, E. A., Endenburg, N., Wijnker, J. J., Rodenburg, T. B., Van Lith, H. A., & Vermetten, E. (2019). The Translation and Validation of the Dutch Monash Dog–Owner Relationship Scale (MDORS). Animals, 9(5), 249. https://doi.org/10.3390/ani9050249
Herwijnen, I. R., van der Borg, J. A. M., Naguib, M., & Beerda, B. (2018). The existence of parenting styles in the owner-dog relationship. PLOS ONE, 13(2), e0193471. https://doi.org/10.1371/journal.pone.0193471
Turcsán, B., Wallis, L., Virányi, Z., Range, F., Müller, C. A., Huber, L., & Riemer, S. (2018) Personality traits in companion dogs—Results from the VIDOPET. PLOS ONE, 13(4), e0195448. https://doi.org/10.1371/journal.pone.0195448
Pongrácz, P., Rieger, G., Vékony, K. (2021) Grumpy Dogs Are Smart Learners—the Association between Dog–Owner Relationship and Dogs’ Performance in a Social Learning Task. Animals, 11, doi:10.3390/ani11040961
Line 290-299: concerning decreased begging behavior, is this finding consistent with other studies involving canine frustration or cognitive decline with age?
RESPONSE: Thank you for the interesting question. There is evidence that stranger-related social behaviours decrease with age (Bognár et al., 2021, Wallis et al., 2014), and also the juvenile origin of specific “begging” behaviours.
We added the following text to the Discussion (lines 378-383).
“Our results are in line with previous findings of age-related decrease in stranger directed contact behaviours (greeting and playfulness [50]) and social attention [51]. There is also some evidence that ‘begging’ behaviour is more pronounced in lower ranking and in juvenile dogs, too. Cafazzo et al. [32] suggested that in subordinate dogs, the more frequent ‘begging’ displays could have their evolutionary origin in the food-requesting behaviour of juvenile individuals.”
Cafazzo, S., Valsecchi, P., Bonanni, R., & Natoli, E. (2010). Dominance in relation to age, sex, and competitive contexts in a group of free-ranging domestic dogs. Behavioral Ecology, 21(3), 443-455.
Bognár, Z., Szabó, D., Deés, A., Kubinyi, E. (2021) Shorter Headed Dogs, Visually Cooperative Breeds, Younger and Playful Dogs Form Eye Contact Faster with an Unfamiliar Human. Sci Rep, 11, 9293, doi:10.1038/s41598-021-88702-w
Wallis, L.J., Range, F., Müller, C.A., Serisier, S., Huber, L., Virányi, Z. (2014) Lifespan Development of Attentiveness in Domestic Dogs: Drawing Parallels with Humans. Frontiers in Psychology 2014, 5, doi:10.3389/fpsyg.2014.00071.
Line 304-307: the association between agonistic scores and determined behavior is very interesting, but did the authors find any confounding factors, such as breed type or prior training. Was it tested?
RESPONSE: Thank you for pointing out the possibility of other confounding factors. Unfortunately, we did not have a large enough breed variety or large enough sample size to investigate breed effect, and we also did not have an a priori hypothesis for associations between rank and breed. Wallis et al., (2020) found in a questionnaire study that according to owners’ perception, more ‘trainable’ dogs also likely were the higher ranked ones. Unfortunately, with a sample size of 39, we have to carefully select what analyses to perform and avoid interaction-analyses altogether, as we could show no real effect.
We have added this text to the limitations (Lines 410-420):
“Another limitation of the study is its low sample size, which made it impossible to investigate some possible confounding factors, such as training level and breed (as there were only one or two dogs from one breed); or to find possible interactions between variables. Future research on frustration should focus on these. Our test population was basically a convenience sample of dog breeds that also included mixed-breeds. As we did not aim to find breed-related effects, a balanced convenience sample can be a suitable solution [55]. Although functional dog breeding could result in differences in the persistence and reliance on humans between cooperative and independent working dog breeds [13], and there are indications that genetic distance from the wolf-like ancestor can also affect persistence and human-directed gazing in dogs [56], we are not aware of any studies that would discover an association between dog breeds and rank position.”
Line 319-330: the results showing differences in experimenter-directed behaviors between dominant and subordinate dogs could be strengthened by further qualitative descriptions of what these behaviors looked like in practice.
RESPONSE: Table 1 contains the detailed description of all the coded behavioral variables.
Line 371-376: the authors acknowledge the exclusion of formal rank from the analysis, but this limitation should be more thoroughly addressed, particularly regarding how it might have influenced the findings.
RESPONSE: Thank you for this suggestion. We have further elaborated this issue in the limitations of the study (lines 406-409). The added text reads like this:
“The exclusion of formal rank may lead to an overemphasis on overt dominance behaviours and situations of conflict (i.e., resource competition and group defence), oversimplifying the relationship between rank and behaviour.”
Reviewer 3 Report
Comments and Suggestions for Authors
Dear Authors,
Thank you for the opportunity to review your manuscript, "Mildly infuriating—rank-related differences in dogs’ behaviours in frustrating situations."
While your study investigates an interesting topic with a sound methodology, I believe the manuscript requires significant revisions before it is ready for publication. I had to reread several sections to fully understand the research and the objectives, as the presentation and clarity currently hinder the reader’s experience.
There are several spelling and formatting issues throughout the manuscript that should be addressed. For example, on line 25, "the" is used instead of "they," there is a double full stop on line 77, and missing or doubled brackets can be seen on line 69. Some references, such as those in line 93, should also be placed at the end of sentences. A thorough proofreading will resolve these issues and improve the overall flow of the text.
The introduction would benefit from being more concise and structured more logically. Some key terms central to the study, such as ‘dominant’ and ‘subordinate,’ should be clearly defined in the introduction, as they are critical to understanding the experiments. Including these terms in the keywords would also be helpful. Additionally, the research question and hypotheses need to be stated more explicitly.
I recommend avoiding the use of first-person language to maintain a formal tone. Ensure that abbreviations, such as the one on line 139, are defined before their first use, and remove any unnecessary commas.
In the methods section, please confirm that ethical approvals were obtained. It would also be helpful to include abbreviations for each test object in the figures to improve clarity. Table titles should be concise, with additional details provided in the notes section. The current table formatting is somewhat difficult to read, so simplifying the numbers and improving the layout would enhance readability.
The discussion needs more focus and depth. It currently reads like an extension of the results section rather than a critical interpretation of the findings. The limitations section is unclear and should more explicitly address why the identified limitation is significant. Additionally, the practical implications of your study should be expanded upon, along with suggestions for future research.
The conclusion requires further development as well. It should provide a clear summary of the main findings, highlight their significance, and offer insight into the broader implications of the study. This section feels underdeveloped and could benefit from a stronger emphasis on your research’s contribution to the field.
In general, I recommend thorough proofreading for spelling, formatting, and sentence structure. Some sentences are overly long and complex, making them difficult to follow. Simplifying these will improve overall readability and comprehension.
Kind regards
Author Response
RESPONSES TO REVIEWER 3
Dear Authors,
Thank you for the opportunity to review your manuscript, "Mildly infuriating—rank-related differences in dogs’ behaviours in frustrating situations."
While your study investigates an interesting topic with a sound methodology, I believe the manuscript requires significant revisions before it is ready for publication. I had to reread several sections to fully understand the research and the objectives, as the presentation and clarity currently hinder the reader’s experience.
There are several spelling and formatting issues throughout the manuscript that should be addressed. For example, on line 25, "the" is used instead of "they," there is a double full stop on line 77, and missing or doubled brackets can be seen on line 69. Some references, such as those in line 93, should also be placed at the end of sentences. A thorough proofreading will resolve these issues and improve the overall flow of the text.
RESPONSE: We are thankful to the Reviewer’s suggestions and efforts to help as in the improvement of our work. We did our best to correct all the formatting and grammatical problems of the manuscript. Thank you for pointing out these typos, we have corrected them.
The introduction would benefit from being more concise and structured more logically. Some key terms central to the study, such as ‘dominant’ and ‘subordinate,’ should be clearly defined in the introduction, as they are critical to understanding the experiments. Including these terms in the keywords would also be helpful. Additionally, the research question and hypotheses need to be stated more explicitly.
RESPONSE: We rewrote the aims/hypotheses/predictions section, hopefully now it gives a clearer outline for the study. We added a section where we briefly explain the definition of dominant and subordinate dogs according to the ethological approach (Lines 102-109). The new text (in blue) reads like this:
“Position of an individual within the hierarchy develops through social interactions [31] and may even change dynamically during the life of the dog [32]. Following the well-established ethological definitions [33], a ‘dominant dog’ is the individual who regularly prevails over the other cohabitant or group member dog (the ‘subordinate’) in dyadic competitive events, and as consequence, gains priority to resources. There are indications that among dogs, a well-established hierarchy is mostly maintained by signals of submission from the subordinate dogs rather than by repeated displays or acts of aggression [32].”
I recommend avoiding the use of first-person language to maintain a formal tone.
RESPONSE: We changed the text to third person at several places; however, we also wanted to avoid the excessive usage of passive sentences, thus we kept first-person structures as well (e.g., “We hypothesized…” “We tested…”). We leave it to the decision of the Editors which would be the more suitable for the Journal.
Ensure that abbreviations, such as the one on line 139, are defined before their first use, and remove any unnecessary commas.
RESPONSE: We have added the full name of the questionnaire and removed the comma before the citation.
In the methods section, please confirm that ethical approvals were obtained.
RESPONSE: Thank you for highlighting this important detail. We naturally received the ethical approval for our study from the Institutional Ethics Committee, and included it to the manuscript. It is the journal’s default format to have ethical approvals and related information separately at the end of the manuscript. You can find this information in Lines 456-463.
It would also be helpful to include abbreviations for each test object in the figures to improve clarity.
RESPONSE: Unfortunately, we weren’t sure which Figures the Reviewer meant here. Now we added explanations of the abbreviated terms in the case of Figure 3.
Table titles should be concise, with additional details provided in the notes section.
RESPONSE: Thank you for the suggestion, we have modified Table 2 and 3 to have a simpler title and added more notes.
The current table formatting is somewhat difficult to read, so simplifying the numbers and improving the layout would enhance readability.
RESPONSE: We limited the decimals to four in the case of the p-values in Tables 2 and 3, and in the case of item-loadings in Table 4, and we hope this provides a better outlay now.
The discussion needs more focus and depth. It currently reads like an extension of the results section rather than a critical interpretation of the findings. The limitations section is unclear and should more explicitly address why the identified limitation is significant. Additionally, the practical implications of your study should be expanded upon, along with suggestions for future research.
RESPONSE: Thank you for the encouragement, we added new sections to the Discussion, where we drew broader parallels between our results and the existing literature. We also considerably extended the limitations section with additional items and possible implications (Lines 401-420).
The conclusion requires further development as well. It should provide a clear summary of the main findings, highlight their significance, and offer insight into the broader implications of the study. This section feels underdeveloped and could benefit from a stronger emphasis on your research’s contribution to the field.
RESPONSE: We extended the Conclusions with some intriguing new research ideas that could capitalize on the fresh findings from an applied aspect (Lines 433-444):
“As for the future, especially from the aspect of applied research, our results provide an interesting starting point for new experiments. It could be tested whether dominant or subordinate dogs would perform better in such scenarios, where the participation of a human handler is either unlikely, or just the opposite, it is necessary. Based on the findings of this paper, we would predict that subordinate dogs would probably be better suited to such tasks, where they do not need to rely constantly on human guidance. However, dominant dogs probably would perform better when they need to work in close cooperation with their handler. Future research on social behaviours and abilities of companion dogs should preferably also take into consideration the possible influence of the dog living in a single dog versus multi-dog household, and the rank of the dog in the case of the latter, as it seems to not only affect intraspecific behaviours, but behaviours in a broader range of social situations.”
In general, I recommend thorough proofreading for spelling, formatting, and sentence structure. Some sentences are overly long and complex, making them difficult to follow. Simplifying these will improve overall readability and comprehension.
RESPONSE: Thank you for the supportive criticism, we hope that the revised paper fits better to the publishing standards of the Journal.
Round 2
Reviewer 3 Report
Comments and Suggestions for Authors
Dear Authors,
I hope this email finds you well.
Thank you for taking the time to address the necessary changes—best of luck with your publication.
Sincerely,
Leena